# Combining Action Observation Treatment with a Brain–Computer Interface System: Perspectives on Neurorehabilitation

**DOI:** 10.3390/s21248504

**Published:** 2021-12-20

**Authors:** Fabio Rossi, Federica Savi, Andrea Prestia, Andrea Mongardi, Danilo Demarchi, Giovanni Buccino

**Affiliations:** 1Department of Electronics and Telecommunications, Politecnico di Torino, 10129 Turin, Italy; fabio.rossi@polito.it (F.R.); andrea.prestia@polito.it (A.P.); andrea.mongardi@polito.it (A.M.); danilo.demarchi@polito.it (D.D.); 2Fondazione Don Carlo Gnocchi, Piazzale dei Servi 3, 43100 Parma, Italy; fesavi@dongnocchi.it; 3Division of Neuroscience, IRCCS San Raffaele Scientific Institute, University San Raffaele, Via Olgettina 60, 20132 Milan, Italy

**Keywords:** action observation treatment, brain–computer interface, functional electrical stimulation, mirror mechanism, neurorehabilitation

## Abstract

Action observation treatment (AOT) exploits a neurophysiological mechanism, matching an observed action on the neural substrates where that action is motorically represented. This mechanism is also known as mirror mechanism. In a typical AOT session, one can distinguish an observation phase and an execution phase. During the observation phase, the patient observes a daily action and soon after, during the execution phase, he/she is asked to perform the observed action at the best of his/her ability. Indeed, the execution phase may sometimes be difficult for those patients where motor impairment is severe. Although, in the current practice, the physiotherapist does not intervene on the quality of the execution phase, here, we propose a stimulation system based on neurophysiological parameters. This perspective article focuses on the possibility to combine AOT with a brain–computer interface system (BCI) that stimulates upper limb muscles, thus facilitating the execution of actions during a rehabilitation session. Combining a rehabilitation tool that is well-grounded in neurophysiology with a stimulation system, such as the one proposed, may improve the efficacy of AOT in the treatment of severe neurological patients, including stroke patients, Parkinson’s disease patients, and children with cerebral palsy.

## 1. Towards Translational, Evidence-Based Approaches in Neurorehabilitation

There is increasing demand in neurorehabilitation for approaches aimed at helping patients to recover functions, and improve their capacity to face daily activities and social interactions.

Following this general aim, any approach in neurorehabilitation should have at least three main features [1,2]:Firstly, it should be evidence based. The efficacy of any rehabilitation practice should be supported by the results merging from randomized controlled studies or clinical trials, comparing a specific approach with a control condition.A neurorehabilitation approach needs to be grounded in neurophysiology: every approach should have its theoretical background in physiology principles and mechanisms. For example, when speaking about motor recovery, the terrific advance of knowledge regarding the organization and functions of the motor system coming from basic neuroscience should be taken into account. All approaches should consider neuroscientific studies to transfer knowledge in clinical practice.Any approach in neurorehabilitation should also aim at the recovery of functions and, as a consequence, of the capacity of patients to interact with the environment and other people, as assumed when considering health not only as the condition in which individuals are free from diseases [3]. It is worth stressing that in many cases, physiotherapists focus on ways to circumvent functional deficits, suggesting alternative strategies in order to allow patients to face daily activities. This attitude leads to a compensation or a reeducation of functions, rather than a cure for them through remediation. In contrast with this rather diffuse attitude, we believe rehabilitative tools should aim at restoring the neural structures whose damage caused the impaired functions, or activating supplementary or related pathways that may perform the original functions.

There are indeed well-known and widespread approaches in neurorehabilitation that fully fit these criteria. One example is constraint-induced movement therapy (CIMT). By means of CIMT, patients promote the use of the more-affected extremities, while restricting the use of the less-affected limb. More specifically, they perform repetitive goal-oriented tasks with the affected limb 6–7 h per day, while wearing a mitt to avoid the use of the less-affected arm for 90% of their waking hours. However, the feasibility of this practice is questioned because patients experience difficulty in complying, due to the fatigue that results from long periods of practice and restraint [4]. CIMT finds application in patients with acute and chronic stroke [4,5], and in children with cerebral palsy, where a gentler version is used than for adults [6]. CIMT induces brain plastic changes, thus leading to a functional reorganization of sensorimotor circuits in monkeys [7].

Another example is the so-called mirror therapy. In this treatment, a mirror is positioned on a table in the patient’s midsagittal plane. Patients place the affected limb behind a mirror and the unaffected limb in front of the mirror. In this way, when they move the unaffected extremity, they get the feeling (a sort of proprioceptive and kinesthetic sensation) to move the affected limb. Treatment normally lasts 10 consecutive weekdays. This approach is effective to relieve phantom pain in arm amputees, as well as in the recovery of upper limbs in chronic stroke patients [8,9]. More recently, it has been effectively applied in the rehabilitation of children with cerebral palsy [10]. The authors that originally proposed mirror therapy as a rehabilitative tool have also suggested that the underlying neurophysiological basis is the mirror mechanism (see below).

Motor imagery is the capacity of individuals to imagine seeing themselves performing a particular action, while generating the kinesthetic experience of the movement, in the absence of motor output [11]. This mental capacity has been practiced for years in sport training and neurorehabilitation. An early study showed that motor imagery may improve posture in elderly people [12]. More recently, using motor imagery, good results have been obtained in the recovery of stroke patients [13,14]. Motor imagery has also revealed a promising approach in Parkinson’s disease patients [15]. It is acknowledged that during motor imagery, similar motor representations are re-enacted as during action execution and action observation, thus suggesting a potential “equivalence” of motor execution, motor imagery, and action observation [16,17]. Hence, all these functions have the potential to serve as rehabilitative tools. However, some studies have suggested that patients with damage to specific brain structures, including the parietal and frontal lobes, lose the capacity to imagine motorically [18]. For this reason, in these patients the use of motor imagery (MI) as a rehabilitative tool can be questionable.

## 2. Action Observation Treatment and Its Efficacy in Clinical Practice

There is increasing empirical evidence that when individuals observe actions performed by other people, they automatically activate the brain neural structures responsible for the actual execution of the seen action [19]. This capacity to match an observed action on its motor representation is defined as the mirror mechanism. At the neural level, this mechanism is most likely encoded by mirror neurons [20]. These neurons were first found in monkey premotor cortexes, and are discharged during both the execution and observation of an action, typically directed towards an object. The existence of a mirror mechanism can explain why healthy adults can learn and/or improve the performance of a specific motor task by observing other people executing it [21,22,23,24,25,26].

Action observation treatment (AOT) is a rehabilitation tool exploiting the mirror mechanism and its potential role in learning and/or re-learning specific motor tasks for the recovery of motor functions, in both neurological and non-neurological patients [27]. During a typical AOT rehabilitation session, patients practice a single daily action (e.g., in one session they practice having an espresso, in another they wash their hands, and so on). Practiced actions are chosen among those of great relevance in everyday life. An AOT rehabilitation session consists of an observation phase and an execution phase. During the observation phase, the patient sits in front of a computer screen and has to carefully observe a video clip, depicting the daily action to be practiced in that session. The presented action can be divided into four motor acts. By motor acts, we mean the different motor segments in which a daily action can be divided. For example, washing one’s hands can be divided into the following motor acts: (i) applying soap on wet hands; (ii) scrubbing hands for a few seconds; (iii) rinsing hands under running water; (iv) drying them with a towel. Each motor act is typically seen for 3 min, so that the whole duration of a video clip depicting the specific daily action is 12 min. In the video, each motor act is performed by both an actor and an actress, and is seen from different perspectives (frontal or lateral view, in foreground and background). This is to make the videos more interesting and relevant, from an attentional point of view. The importance of showing actions from different perspectives is supported by a monkey study [28], where the authors found that most mirror neurons are sensitive to different visual perspectives. Moreover, in a recent review, the authors suggested that different perspectives may be helpful for different motors skills. For example, first-person perspective could be helpful for manual dexterity tasks (showing more visual cues, which are important to perform the task successfully) and for the embodiment of the movements. On the other hand, third-person could be best for more gross motor tasks [29]. After observing each motor act for 3 min (observation phase), patients move to the execution phase, when they must perform for 2 min what they have just seen. Although patients are requested to execute the observed motor act at the best of their ability, they are informed that the focus of the treatment is on the observation of the action, not its execution. During the execution phase, objects used in the video clip are provided at hand, in order to allow patients to execute the observed action in a realistic context. Note that objects are known to automatically trigger the most useful motor representations that can be acted upon them, thus further contributing to the reorganization of the motor system [30,31,32]. Moreover, there is empirical evidence that the recruitment of the motor system is fine-tuned with the motorically relevant features of an observed object [33].

A typical AOT rehabilitation session lasts half an hour. The physiotherapist explains the task to the patient for a few minutes (e.g., look carefully at the movie, pay attention to all the details of presented actions) to motivate him/her. Then, 12 min are devoted to the observation phase (3 min for each presented motor act) and finally 8 min to the execution phase (2 min for each motor act). For adults, the rehabilitation program with AOT includes the practice of 20 daily actions, and takes 4 weeks (5 working days a week).

Thus far, AOT has been used in the rehabilitation of patients suffering from chronic ischemic stroke (more than six months after the acute event), in Parkinson’s disease patients, in children with cerebral palsy, and in orthopedic patients undergoing surgery of the hip or knee (for a comprehensive review see [34]). In a pivotal, randomized controlled study of patients with chronic ischemic stroke in the territory of the middle cerebral artery [35], AOT was applied to recover upper limb motor functions. Patients in the control group had to observe video clips with no specific motor content. The functional outcome of both groups was assessed by means of the stroke impact scale, the Wolf motor function test, and the Frenchay arm test. After treatment, the patients undergoing AOT achieved better scores than patients in the control group, in all scales. An improvement was still evident at the two-month follow-up. Moreover, a functional magnetic resonance imaging (fMRI) study, carried out during an independent motor task, namely free object manipulation, before and after AOT treatment, showed a significant increase in the activation of brain areas endowed with a mirror mechanism in the AOT patients after treatment.

The effectiveness of AOT has been investigated also in PD patients, with the aim to complement drug treatment in these patients [36]. Similar to stroke patients, participants in the case group observed videos depicting everyday life actions, including postural actions and walking, whereas those in the control group observed movies devoid of any specific motor content. After treatment, patients in the case group scored better than controls on two functional scales: the unified Parkinson’s disease rating scale (UPDRS) and the functional independence measure (FIM). AOT has also been successfully applied in remediation of freezing of gait in Parkinson’s disease (PD) patients [37]. These findings can be explained by the evidence that the basal ganglia, the neural structures most affected in PD, are heavily connected with the brain areas involved in the mirror mechanism, as well as in motor learning and motor planning [38]. It is most likely that in PD patients, AOT may contribute to reorganizing the normal loop circuits connecting the motor cortex with the basal ganglia [39]. In PD patients, during action observation, changes in beta oscillatory activity of the subthalamic nucleus occur, similar to those recorded over the motor cortex, thus suggesting that basal ganglia may play a role in the mirror mechanism [40]. Furthermore, it is well-known that PD patients improve their motor performances when they obtain an external cue [41]. One could argue that the visual stimulation provided by means of AOT represents an appropriate cue to start and execute several daily actions in PD patients. Moreover, in addition to cueing actions, AOT could also reinforce the normal connections in the brain between the prefrontal and premotor cortex. These connections play a fundamental role in paying attention to and performing actions in a convenient context in healthy people, and appear compromised in PD patients [42].

AOT has been applied for the recovery of upper limb motor functions in children with cerebral palsy [43]. In this study, children in the case group observed daily actions appropriate for their age, whereas children in the control group observed documentaries with no specific motor content. A functional evaluation with the Melbourne assessment scale of upper limb motor functions showed that children undergoing AOT scored significantly better than the controls after treatment. Other authors have confirmed the effectiveness of AOT in the recovery of upper limb motor functions in children with cerebral palsy [44,45]. Moreover, in a recent fMRI study [46], children treated with AOT showed stronger activation in brain areas subserving the execution of actions, and implying the use of objects. These findings support the notion that AOT contributes to the reorganization of brain circuits subserving the impaired function, rather than activating supplementary or vicariating circuits.

Interestingly, AOT may also promote motor recovery in post-surgical orthopedic patients for hip fractures or elective hip or knee replacements [47]. In a pivotal study, patients in the experimental group observed video clips depicting daily actions performed with lower limbs and subsequently imitated them. Patients in the control group observed video clips with no motor content, and then executed the same actions as the AOT group. Two functional scales (FIM and Tinetti scale) were used to assess the functional outcome. After treatment, patients in the AOT group scored better than patients in the control group in both functional scales. Moreover, patients in the case group were prescribed a walker less frequently than controls at discharge. These findings suggest that AOT is an effective adjunct to conventional therapy in the rehabilitation of post-surgical orthopedic patients. In more general terms, the findings of this study suggest a top-down effect in neurorehabilitation, showing that the reorganization of motor representations at central level, most likely occurring during AOT, may positively affect performance, even when the skeletal structures to implement actions are affected.

Finally, it is worth underlining that AOT has been tested as a tool on the rehabilitation of aphasic patients. A case report study supports the notion that the observation and execution of actions can improve the recall of action words in patients with a selective deficit for verb retrieval [48].

For the aim of the present study, it is relevant to underline that AOT is a flexible tool in neurorehabilitation: in fact, actions trained during rehabilitation sessions can be adapted to the real needs of patients. For example, patients impaired in performing distal hand/arm actions (i.e., grasping, manipulating) should practice these motor tasks, whereas those presenting with impairment of proximal arm actions (i.e., reaching objects, coding objects in space) should focus on these motor tasks. Moreover, this individual training program has the potential to be applied in a tele-rehabilitation setting. Tele-rehabilitation exploits telecommunication devices to provide evaluation, distance support and eventually rehabilitation for patients living at home. Despite the fact that so far tele-rehabilitation has been mainly applied in the rehabilitation of stroke patients [49,50,51], it has the potential to meet the rehabilitation needs of most patients, including neurological and non-neurological patients. Just for example, in the field of neurology, a recent review suggests the potential use of tele-rehabilitation for patients with multiple sclerosis [52], whereas in the field of internal medicine the effectiveness of tele-rehabilitation for patients with chronic respiratory disease has also been assessed [53].

Most current studies have applied AOT in conventional rehabilitation settings (at hospitals or in rehabilitation centers). However, since AOT focuses on the delivery of visual stimuli depicting actions, it can also be easily applied at patients’ home in a tele-rehabilitation setting, under remote control if necessary or appropriate. Caregivers and patients can be easily trained to its use, and they can be provided with all the necessary materials and requisites. Two recent studies support the use of AOT in a tele-rehabilitation setting in children [54,55]. Applying AOT in patient’s home has advantages, when patients must follow a long-term rehabilitation program and/or when they have difficulty in joining rehabilitation centers. The current health emergency due to the COVID-19 pandemic has brought further the use of home-based rehabilitation strategies; in this context, tele-rehabilitation can help to stop the spread of contagion, while assuring patients with motor impairment the possibility of following a regular rehabilitation program.

## 3. Combining AOT with a Brain–Computer Interface to Improve the Actual Motor Execution of Patients

So far, during the execution phase, patients have been requested to execute the observed motor act to the best of their abilities. Indeed, in a typical AOT protocol, physiotherapists do not intervene to improve patients’ performance, but they ask patients to pay attention to all details of the observed actions, and try to motivate them in the execution phase, when all objects presented in the videoclips are at hand, so that patients can act upon them as they would in everyday life contexts. However, in this study, we propose a stimulation system that should help patients, especially those who are not able to reproduce the observed motor act, or are discouraged from implementing the performance in a physiological manner by an inadequate execution.

Among the therapies involving electrical currents, Functional Electrical Stimulation (FES) induces muscle contraction by exciting the axonal fibers, which innervate the muscular belly [56,57]. The FES low-energy pulses are applied on the skin’s surface, using superficial electrodes, making the approach entirely non-invasive and comfortable [58]. Multi-session FES therapy has been demonstrated to restore motor task functionalities, promote blood circulation, and prevent muscle atrophy, resulting in a significant improvement of daily life for people suffering from neurological disorders [59].

An active control of the FES application could be implemented by monitoring and processing the neurophysiological signals that are the most descriptive of the planning and execution of the movement. Indeed, by assessing the electrical activity of the central nervous system at certain brain regions (e.g., premotor cortex, primary motor cortex), which is known as ElectroEncephaloGraphy (EEG), or by supervising the contraction of the skeletal muscles, represented by the ElectroMyoGraphic (sEMG) signal, bio-mimetic FES-controlled systems have been realized [60,61,62].

Considering the central role of EEG in analyzing the brain response to AOT [63,64,65], a brain–computer interface (BCI) could be developed to control the FES application according to a specific EEG activity [66,67]. In particular, depending on the subject possibilities, the control could be based on both motor imagery (MI) [68,69] and movement execution (ME) approaches, which have been demonstrated to activate almost the same cortical regions [17]. Either way, similarly to AO, an activity change of the mu and beta EEG waves, known as event-related desynchronization (ERD), is induced [70]. However, considering the difficulties related to MI (e.g., long training is required to command the BCI, and it is impossible to assess the correctness of mental training [27,71]), ME has been identified as a more reliable technique for the implementation of the FES control, considering that it includes both motion preparation and actuation phases, thus covering the whole execution process. Although other physiological signals could be used as BCI inputs (e.g., visual stimuli [72]), AO, MI, and ME approaches have been proven to be more beneficial in neurorehabilitation, because the combination of simultaneous cortical activity and muscle stimulation significantly promotes neuroplasticity [73,74].

A representative block-scheme of the architecture of a typical EEG-based BCI-FES system is represented in Figure 1. The detection of the EEG signal could be easily assessed by using standard 32–64 channels EEG caps, which provide the enhanced electrode positions of the International 10–20 systems. Raw EEG signals require the application of a band-pass filter (1 Hz–40 Hz), in order to remove the DC offset and higher frequency interferences (e.g., 50 Hz–60 Hz power line) [70,75]. Depending on the position of the sensing EEG channels, additional signal artifacts (e.g., heartbeat, blinking) should be taken into account [76,77]. After the digitization of the EEG, the power of the signal related to the frequency bands of mu (8 Hz–13 Hz) and beta waves (16 Hz–31 Hz) [70] are estimated, using the well-known mathematical operators of the fast Fourier transform (FFT) or the Wavelet Transform (WT) [75]. A proper ERD identification can be performed by employing machine learning algorithms based on, for example, the neural network (NN) or support vector machines (SVM) classifiers, sometimes combined with common spatial patterns (CSP) for feature extraction [68,75]. At the last stage, the FES application is efficiently controlled depending on the classifier’s output, which discerns between no movement preparation (no need for FES) and volitional action (starting FES).

In this context, the modulation of the FES parameters based on the EEG signal remain an open scenario. Indeed, many works [66,67,78] apply the stimulation with a strict on/off mechanism, without including any modulation of the features of the FES pulses (e.g., amplitude, width). Among the reasons for this is that the EEG represents what is happening at the central nervous system during a specific task only macroscopically; therefore, this makes it challenging to deconstruct this information into the functional data at a peripheral level. As a clear example, we can consider a complex and synergic action like the reaching of an object: the EEG signal provides high-level information about the preparation and execution of the movement (through the analysis of mu and beta waves) but does not allow the discrimination of which muscle is activated (and in which way) to perform the action. For this reason, since the FES operates at the peripheral level by directly stimulating the muscle fibers, a constant per muscle update of the FES parameters could not be easily obtained.

A more efficient FES modulation could be achieved by monitoring the activity of the skeletal muscles involved in the desired movement. The acquisition of the sEMG signal is performed using non-invasive electrodes placed on the skin above the muscle whose activity must be analyzed. The main information content of sEMG is contained in the 50 Hz–150 Hz band, and, from a high-level perspective, its conditioning involves a high-pass filter to remove the DC offset and motion artifacts, one or more amplification stages, and a low-pass filter for antialiasing purposes and high-frequency noise suppression [79,80]. Methods to extract useful features about muscle activity include standard practices, such as sEMG envelope extraction [81,82], and other quantitative evaluators, like the average threshold crossing event-driven approach [83], which are more suitable for low-power and wearable systems [84,85].

In the system proposed in this paper, the sEMG signals will be used as feedback in order to achieve a closed-loop system. In particular, through sEMG it is possible to assess which muscles are contributing to the execution of the movement, so that FES can be applied only to inactive muscles, in order to both encourage voluntary activity and prevent atrophy of unused muscles. Moreover, the sEMG feedback can help to monitor the uprising of muscle fatigue, which is a typical phenomenon related to FES application, because of the opposite order in the muscle fiber recruitment, compared with the physiological one (i.e., the Henneman’s size principle) [86,87]. Among the popular practices for evaluating muscle fatigue, the most employed techniques are the analysis of the down-shift of the median frequency of the sEMG signal spectrum [88], and the analysis of the M-wave (i.e., the FES-evoked EMG response), which provides the quantification of the number of recruited motor units [89,90].

However, combining the sEMG and FES techniques requires to implement appropriate strategies for muscle activity extraction (both the voluntary and the evoked one) and stimulus artifact suppression [91,92], which would otherwise result in saturating the sEMG-sensing electronics [93]. A possible strategy to solve this issue could be to implement at the inputs of the acquisition channel a blanking circuit [93,94], which would have to be appropriately synchronized with the stimulation frequency, and which would also have to discharge the acquisition electrodes to avoid transient responses disturbing the sEMG reading [90,93].

Furthermore, the trajectory evolution of the movement could be monitored by inertial measurement units (IMUs), which, integrating different sensors (i.e., accelerometers, gyroscopes, magnetometers) into a single wearable chip, would allow an accurate kinematic analysis for an object moving in a 3D space [95]. Additionally, since activities of daily living (ADLs) include complex and multi-domain actions (e.g., grasping an object, scratching a shoulder, drinking from a bottle), the presence of sensors to identify when a task is completed could be used to provide feedback to the patient, in order to encourage him/her during the rehabilitation session. Key technologies for implementing this control can range from radio-frequency identification (RFID) [96], through capacitive sensors [97], to body channel communication (BCC) [98,99].

In its early stage of use, the system will be employed in clinics under the supervision of trained medical personnel, requiring a full sensor configuration to obtain robust results in a short time; thus, not burdening the finances of the medical structure. In a second phase, for patients with a neurosensory involvement already established during earlier clinical sessions, or for patients who are uncomfortable in repeatedly getting to the hospital, the system could be used at home for long term rehabilitation in a reduced configuration, in order to ease sensor placement by the patient or the caregiver. In particular, since the main FES-pattern modulation is performed by sEMG acquisition, the EEG and the BCC hardware can be removed from the overall system, thus reducing the physical dimensions of the home equipment.

In our main configuration, as depicted Figure 2, each patient will be equipped with a wearable helmet for the acquisition of the EEG rhythm, some muscular inertial units, which embed sEMG and motion tracking together, an electro-stimulator for the application of FES, and the control software (which could be installed by the user on his/her own computer or, if necessary, on a provided one). The system will receive, via Bluetooth low-energy (BLE) communications, the data from each module, process them, and appropriately modulate the FES parameters (e.g., pulses amplitude and width, stimulation frequency). Furthermore, the software features a graphical user interface, to ease the communication with both the physiotherapist (e.g., session progression, sensors data) and the patient (e.g., displayed messages), and plays the videoclips needed in the observation phase. However, if the patient subjects himself/herself to AOT at home, a properly educated caregiver must be involved, in order to assist him/her during the setup of each sub-device (e.g., ensuring proper electrode placement) and its connection to the central processing unit.

The proposed system’s final aim is to effectively make the subject execute the desired action. Indeed, in the main case scenario, the devices will be activated at the beginning of each execution phase, in order to monitor the physiological signals, and the FES will take action only if the patient attempts to perform the movement without succeeding, in order to encourage voluntary muscle activity. Alternatively, if the EEG monitoring is omitted, the FES onset can be remotely triggered by the therapist, or the patient can automatically follow the task indication.

In particular, as detailed in Figure 3, the main scenario includes the sensors being continuously monitored to obtain useful information about the patient state, and consequently performing different feature extractions:EEG activity is analyzed considering its power density in the beta and mu bands, aiming to detect whether an ERD occurs and to define its magnitude;Muscles activity is monitored by the sEMG sensors, distinguishing between each acquisition channel (i.e., different muscular fiber recruitment) and evaluating if the relation among them reflects the physiological behavior;From sEMG, muscular fatigue assessment is performed too, analyzing the M-waves from the different muscles, and considering its degradation over time;Position, angular velocity, and linear acceleration from the different employed IMUs are combined to reconstruct the limb kinematic across space, evaluating if they are consistent with physiological movements.

The extracted features are sent to a data processing unit, which decides whether the FES has to be applied, and how to tune the different channel parameters to make the movement as natural as possible. A machine learning algorithm takes action at this point to automatically analyze the features together. Thus, having stored several previous data acquired from healthy and unhealthy subjects, the system will be able to recognize the conditions of the patient and will instruct the FES module to update the stimulation parameters accordingly. For example, the machine could recognize different ERD magnitudes and increase the FES current, if the acquired value is lower than the one saved as the physiological standard. Alternatively, it could receive unexpected frequency information by the sEMG sensors and decide to vary the pulse width and frequency of the generated stimulation waves, to relax muscle fatigue.

Lastly, if the performed action requires reaching for an object, BCC or RFID sensors can be involved in detecting the effective extent of the movement itself, giving visual feedback to the user. An encouragement message is displayed when no contact is detected, motivating the subject to complete the exercise, and proceed with the routine. At the end, when the target object is touched, positive feedback is represented and rest time is observed, before subsequent execution.

In the home-based scenario, the data acquired by the sensors are automatically saved in the local storage for subsequent analyses. Furthermore, if the clinician would like to verify in real time the progress of the patient, he/she could activate a data streaming towards their facility, in order to monitor the physiological signals acquired during movement execution.

## 4. Conclusions

In the present perspective article, we proposed a novel way to apply action observation treatment (AOT) in clinical practice. More specifically, we proposed the use of a brain–computer interface (BCI) with the aim to stimulate upper limb muscles during the execution phase of AOT, when patients are requested to execute an observed motor act.

Combining AOT with BCI may present advantages for the patients during the execution phase of AOT. BCI may improve the performance of patients, whatever the gravity of their neurological impairment. Considering the neurological diseases where AOT has been widely applied, with this system we propose that we can stimulate motor activity in stroke patients with upper limb paralysis, severe Parkinson’s disease patients or in children with a severe cerebral palsy.

Like AOT alone, the combination of AOT with BCI has the potential to be used in conventional rehabilitation settings, as well as at home, possibly under remote control. In this case, caregivers could receive information on how to apply electrodes for stimulating muscles under the supervision of clinicians, or even without it, when enough experience has been reached. It is noteworthy that this approach could favor an increased awareness of patients and caregivers in the treatment process, and promote an active role in the cure. At the same time, national healthcare systems could spare resources with advantages for the whole of society.

## Figures and Tables

**Figure 1 sensors-21-08504-f001:**
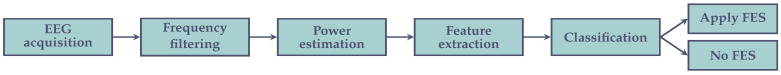
Typical BCI acquisition chain. After acquiring the EEG signal, the frequency components outside the band of interest are filtered out. Then, the power of the signal is estimated, followed by the extraction of its features to be used to classify if FES needs to be applied or not.

**Figure 2 sensors-21-08504-f002:**
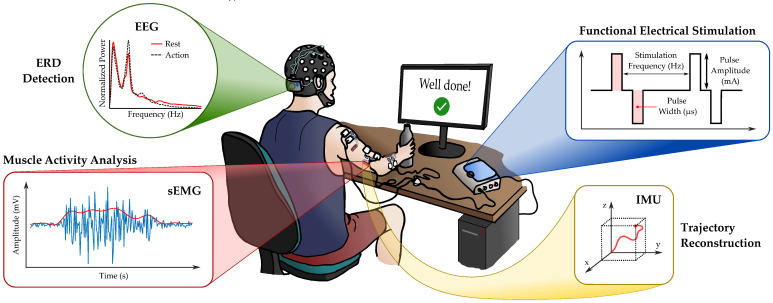
Overview of the system. The patient is instructed to perform the reaching of an object, typically. EEG electrodes are placed on the motor areas and stabilized by a comfortable helmet. sEMG and inertial sensors are placed on the limb of interest, next to the stimulation electrodes. A central unit processes acquired data to activate the FES when the subject needs help to reach the target. A monitor provides feedbacks encouraging the user.

**Figure 3 sensors-21-08504-f003:**
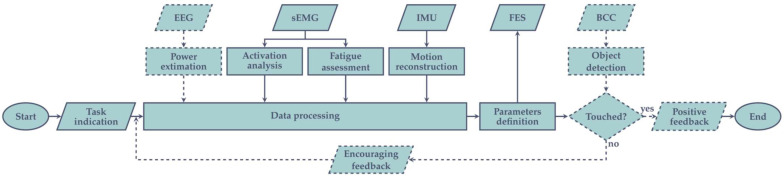
Activity flow of an execution session. After the indication of the task to perform, the subject is required to execute the action. The four sensors are continuously monitored, and for each of them a proper feature is extracted and recorded (e.g., signal power from EEG, limb trajectory from the IMUs). The information obtained by EEG, EMG and IMUs is combined and processed to evaluate how the subject body is reacting, and to decide if the FES must be applied to assist the execution. Therefore, FES parameters are tuned depending on the decision of the processing stage, stabilizing the movement by stimulating one or more muscles, if necessary. Lastly, if the task consists of reaching for an object, the subject is encouraged until the BCC sensors detect the touch of the target.

## Data Availability

Not applicable.

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
