# Peer review of "Combining Action Observation Treatment with a Brain–Computer Interface System: Perspectives on Neurorehabilitation"

_sensors, 2021, doi:10.3390/s21248504_

Round 1

Reviewer 1 Report

Summary

The current review provides an overview of the potential benefits of action observation therapy (AOT) in relation to other therapies typically adopted in clinical populations. In addition, the authors propose a creative and novel use of brain computer interface and functional electrical stimulation to assist the execution phase of AOT. This would certainly be a progressive study proposal that technically and theoretically could accelerate rehabilitation for the clinical populations of interest. However, I do wonder about the feasibility and cost effectiveness of such treatment (e.g., providing this equipment for home-based training) and the training of potential carers to assist in the set up and delivery of this intervention. This review and study proposal will be of great interest to both clinicians and academics; however, I feel some further considerations should be made regarding the feasibility of this intervention if successful.

Lines 58-60 – It could added that these training schedules consisting of only physical practice may be induce greater physical fatigue in individuals in comparison to other approaches promoted by this paper (e.g., AOT – part observation and practice).

Line 60 – Please revise or remove “This for 10-15 consecutive weekdays”.

Lines 75-77 – Please refer more explicitly to the visual modality of imagery to support kinaesthetic generation, in line with Jeannerod (1994). For example, “Motor imagery is the capacity of individuals to imagine seeing themselves performing a particular action while generating the kinaesthetic experience of the movement, in the absence of motor output”.

Jeannerod, M. (1994). The representing brain: Neural correlates of motor intention and imagery. Behavioral and Brain sciences, 17(2), 187-202.

Lines 81-84 – I would use these two citation together with more caution. Indeed, Jeannerod claimed there to be functional equivalence between overt and covert motor simulations; however, the meta-analysis by Hardwick et al. (2018) revealed that these simulation states did not completely overlap, although they do share some neural substrates. Therefore, it might be helpful to rephrase this sentence, for example –

“It is acknowledged that during motor imagery, similar motor representations are re-enacted as during action execution and action observation, thus suggesting a potential “equivalence” of motor execution, motor imagery, and action observation [15], [16].

Lines 94-96 – Please add reference Di Pellegrino, G., Fadiga, L., Fogassi, L., Gallese, V., & Rizzolatti, G. (1992). Understanding motor events: a neurophysiological study. Experimental brain research, 91(1), 176-180.

Line 111 – Add comma before “(iv)”.

Lines 113-118 – Are videos developed to optimise perspective for the tasks? For example, Scott et al. (2020) suggests the first-person perspective during AO could be helpful for manual dexterity tasks (showing more visual cues which are important to performing the task successfully) and for embodiment of the movements. On the other hand, third-person could be best for more gross motor tasks.

Scott, M. W., Wood, G., Holmes, P. S., Williams, J., Marshall, B., & Wright, D. J. (2021). Combined action observation and motor imagery: An intervention to combat the neural and behavioural deficits associated with developmental coordination disorder. Neuroscience & Biobehavioral Reviews.

Lines 122-124 – Typo in sentence, “During the execution phase, objects used in the video clip are provided at hand in order to allow patients to execute the observed action in realistic contexts.”

Section 3 – The study is well thought out and would certainly be of interest to clinicians and researchers. However, I do feel there should be further consideration regarding the feasibility of rolling out such an intervention (if proved beneficial), which involves a lot of equipment and the training of individuals (e.g., carers) to support impaired individuals in the set up and delivery of it.

Reviewer 2 Report

In the paper entitled “Combining Action Observation Treatment with a 1 Brain-Computer Interface System: Perspectives in 2 Neurorehabilitation” the authors describe with details different techniques in rehabilitation based on Action Observation Treatment (AOT). Many references are cited in the text to describe the different techniques. Then, in order to improve these techniques, the authors propose the use of a system based on the record electrophysiological systems to provide feedback for helping the subject rehabilitation.   Some of these signals are based on EEG and EMG activity.

If in the first part of the paper, the authors describe the different techniques based on AOT, in the second part they describe the proposed system, however, there is only a suggestion. There is no implementation, neither experimental session and neither results. In this sense, the paper is only a proposition, and cannot consider as a progress in this topic.   

the authors only introduce a suggestion to improve Action Observation Treatment (AOT). This suggestion is based on the implementation of a system that would record and process EEG and EMG signals. Even if the idea could be interesting, it is not possible to accept a paper based only on a proposition. In order to improve the paper, the authors should implement the system proposed, should carry out experimental tests to validate it, and should compare the obtained results with those obtained using other conventional techniques.

Reviewer 3 Report

In this paper, the author proposed a new method to apply BCI and AOT in clinical practice. They hope to use a Brain-Computer interface to stimulate upper limb muscles during the execution phase of AOT and require the patient to execute an observed motor act, so as to improve the patient's performance. The idea of this article is very novel and I think it can be accepted.
